# Prevention of Gestational Diabetes Mellitus and Gestational Weight Gain Restriction in Overweight/Obese Pregnant Women: A Systematic Review and Network Meta-Analysis

**DOI:** 10.3390/nu14122383

**Published:** 2022-06-09

**Authors:** Shan Wu, Jiani Jin, Kai-Lun Hu, Yiqing Wu, Dan Zhang

**Affiliations:** Key Laboratory of Reproductive Genetics (Ministry of Education), Department of Reproductive Endocrinology, Women’s Hospital, Zhejiang University School of Medicine, Hangzhou 310006, China; 12018530@zju.edu.cn (S.W.); 22018159@zju.edu.cn (J.J.); 11918387@zju.edu.cn (K.-L.H.); wuyiqing@zju.edu.cn (Y.W.)

**Keywords:** gestational diabetes mellitus, gestational weight gain, overweight, obesity

## Abstract

Background: Overweight/obesity is associated with pregnancy-related disorders, such as gestational diabetes mellitus (GDM) and excessive gestational weight gain (GWG). Although multiple interventions have been proposed to prevent GDM and restrict GWG, our knowledge of their comparative efficacy is limited. Objective: To evaluate the effectiveness and identify the optimal intervention strategy to prevent GDM and restrict GWG among overweight/obese pregnant women. Methods: Randomized controlled trials that recruited overweight/obese pregnant women at <20 gestational week were obtained. Predictive and confidence interval plot and surface under the cumulative ranking (SUCRA) were performed using Stata statistical software to determine and compare the efficacy of interventions (diet, physical activity (PA), diet + PA intervention and medication). Results: 23 studies with a total of 8877 participants were eligible for analysis. Our results indicated that although neither PA, diet + PA, diet nor medication intervention could significantly protect overweight/obese women from the development of GDM, there was a trend that PA and diet + PA intervention were preventive factors of GDM. Of these, PA intervention (SUCRA, 82.8%) ranked as the superior strategy, and diet intervention (SUCRA, 19.7%) was the least efficacious regimen. Furthermore, interventions of diet, PA and diet + PA were significantly beneficial for GWG restriction, whereas medication intervention could not restrict GWG. In detail, diet intervention (SUCRA, 19.7%) ranked as the optimal regimen, whilst PA intervention (SUCRA, 62.3%) ranked as the least efficacious regimen. Conclusion: Although none of the interventions could offer remarkable benefit for GDM prevention, interventions of diet, PA and diet + PA were significant factors to restrict GWG. In aggregate, diet + PA intervention seemed the superior choice for the prevention of both GDM and excessive GWG. Registration: PROSPERO CRD42022313542.

## 1. Introduction

Gestational diabetes mellitus (GDM) has for many years been defined as glucose intolerance with the onset or first recognition during pregnancy. Such a definition has serious limitations due to many cases of GDM representing preexisting hyperglycemia [1]. As such, the latest definition of GDM excludes women found to have diabetes by diagnostic criteria applied outside of pregnancy [1]. GDM is one of the most common obstetric complications, with the prevalence varying from 7.5% to 27.0% among different areas, principally depending on different races and diagnostic criteria [2]. GDM is associated with a wide variety of adverse maternal and offspring outcomes. Women with GDM are at higher risks of pre-eclampsia, dystocia, cesarean section, postpartum hemorrhage, and future development of type 2 diabetes mellitus. GDM also contributes to macrosomia, childhood obesity, metabolic syndrome and cardiovascular diseases in the offspring [3]. Overweight and obesity are leading global health burdens and constitute major risk factors of GDM [4]. The proportion of overweight and obese women of reproductive age has been increasing considerably in the last decades. It is estimated that among overweight and obese pregnant women, the risk of developing GDM is more than twofold higher than for non-obese women [5]. Furthermore, the combination of overweight/obesity and GDM could aggravate these adverse outcomes caused by GDM alone [6].

The Institute of Medicine (IOM) has recommended the ideal gestational weight gain (GWG) for pregnant women. According to the IOM guidelines, total GWG should be within 12.5–18 kg, 11.5–16 kg, 7–11.5 kg and 5–9 kg for underweight, normal weight, overweight and obese women, respectively [7]. Excessive GWG increases the risk of GDM, hypertensive disorders of pregnancy, large for gestational age infants, macrosomia, caesarean delivery, and postpartum weight retention, as well as obesity of offspring [7,8,9]. As reported by a systematic review and meta-analysis of more than 1 million pregnant women, 47% of the population exceeded GWG goals [7], and overweight and obese women have the highest prevalence of excessive GWG [10]. Therefore, effective interventions targeting women who are overweight or obese are urgently needed to decrease the risk of GDM, restrict GWG and promote the future health of two generations.

There are substantial randomized controlled trials (RCTs) attempting to reduce the incidence of GDM, including diet, physical activity (PA), combined interventions or medication. However, the results are controversial [11,12]. The key issue of whether GDM could be prevented by interventions remains unanswered. Furthermore, the effects of interventions on limiting GWG are also inconsistent [13,14,15,16]. In addition, individual RCTs focusing on diverse interventions present inconsistent effectiveness [5,17,18], indicating that the efficacy of various strategies might be different. In this regard, few studies compare the effectiveness of multiple interventions, especially in overweight/obese pregnant women. Consequently, it is necessary to determine the optimal intervention among this population, as well as to update the meta-analysis since certain new trials have been conducted recently.

In this network meta-analysis, we aimed to assess the comparative efficacy of different interventions during pregnancy on preventing GDM and restricting GWG among overweight/obese women, and establish the optimal strategy for clinicians.

## 2. Materials and Methods

### 2.1. Data Sources and Search Strategy

This systematic review and network meta-analysis was reported according to the Preferred Reporting Items for Systematic Reviews and Meta-Analyses guidelines (PRISMA) [19]. The study protocol has been registered (registration number: CRD42022313542) with the International Prospective Register of Systematic Reviews (PROSPERO). We searched PubMed, Cochrane Library, Web of Science and Embase from inception to 29 March 2021 for RCTs published in English. The search strategy was constructed following the PICOS tool: (P) Population: overweight/obese pregnant women at <20 gestational weeks; (I) Intervention: diet, PA, lifestyle intervention and medication; (C) Comparator: usual care; (O) Outcomes: the incidence of GDM and GWG, and (S) Study type: RCTs. Detailed search strategies were listed in Appendix A and supplemented with manual searches of the reference lists of included studies and relevant reviews as well as the PubMed option ‘Related Articles’.

### 2.2. Study Selection

Inclusion criteria principally included RCTs that evaluated diet, PA, diet + PA and medication interventions among the overweight/obese population with the measurement of GDM incidence or GWG. In addition, only studies recruiting participants at <20 weeks’ gestation, with singleton pregnancy, and without pre-existing type 1 or type 2 diabetes mellitus were eligible for inclusion. Exclusion criteria were having GDM in the current pregnancy. A wide spectrum of GDM diagnostic criteria were adopted by included trials, such as the criteria defined by the International Association of Diabetes and Pregnancy Study Groups (IADPSG), the World Health Organization (WHO), the American Diabetes Association (ADA), Australasian Diabetes in Pregnancy Society (ADIPS), the Canadian Diabetes Association, and so on. GWG was majorly calculated as the difference between the last measured weight before delivery and pre-pregnancy weight, and the weight first measured in early pregnancy would be used if pre-pregnancy weight was unavailable. The definition of evaluated interventions was as follows: diet intervention included education sessions, specific consultations or additional materials for dietary advices; PA intervention included personalized PA recommendations, supervised aerobic exercise sessions or additional materials for PA advices; Combined diet + PA intervention indicated the combination of the above diet and PA intervention strategies; Medication intervention included all types of oral hypoglycemic agents. Control group meant standard prenatal care (including with/without general nutrition/PA consultations) or placebo tablets. Studies on dietary interventions with only nutrient supplements were excluded. Two reviewers (S.W. and J.J.) independently selected studies based on titles and abstracts; if eligibility was still undetermined, full study texts were reviewed. Discrepancies between reviewers were resolved by consensus with a third researcher (K.-L.H.).

### 2.3. Data Extraction

Two reviewers (S.W. and J.J.) independently reviewed each study and extracted data with pre-designed standardized collection forms. Extracted data include author, publication year, conflict of interest, ethical approval, time frame, country, inclusion and exclusion criteria, sample size, intervention measures and duration, control measures, and outcome reported. When required, corresponding authors of the studies were contacted for data clarification. After data extraction, disagreements or uncertainties were discussed with another researcher (K.-L.H.) until consensus was achieved.

### 2.4. Quality Assessment

Two independent investigators (S.W. and J.J.) assessed study quality using the Cochrane Collaboration’s tool [20]. The assessment for the risk of bias (ROB) addressed seven specific items, including random sequence generation, allocation concealment, blinding of participants and personnel, blinding of outcome assessment, selective reporting, incomplete outcome data and other sources of bias. Each item was assessed, and ROB was rated as low risk, high risk or unclear. Any disagreement was resolved by discussion with another researcher (K.-L.H.). Subsequently, trials were categorized into three levels of ROB according to the number of components for potential high ROB existence: high risk (*n* ≥ 5), moderate risk (*n* = 3–4) and low risk (*n* ≤ 2).

### 2.5. Statistical Analysis

Stata 16.1 software (StataCorp, College Station, TX, USA) was used to analyze the extracted data. We performed pairwise meta-analyses with a random-effects model in Stata if direct data were available. Using the results obtained from the network meta-analysis, we presented a summary of treatment effects as risk ratios (RRs) and mean difference (MD) with 95% CI to facilitate the interpretation of the results in terms of the magnitude of heterogeneity. In order to assess agreement between direct and indirect evidence within the network, the inconsistency plot was performed by using the ‘ifplot’ function of Stata. We also presented the predictive intervals by the ‘intervalplot’ function of Stata, which provide an interval in which the estimate of a future study might be expected.

The ‘networkplot’ function of Stata was employed to construct network plots to illustrate the geometry of evaluated interventions. In the plots generated, nodes represent the different interventions and the control condition of usual care, and lines connecting the nodes represented the direct head-to-head comparisons between interventions. In order to assess publication bias, a network funnel plot was generated and visually inspected using the criterion of symmetry. To evaluate the contribution of different direct comparisons to this network meta-analysis (NMA), the ‘netweight’ function of Stata was utilized to draw the contribution plot which helps in identifying the large and small contribution of evidence in a network. Probability values were summarized and are reported as the surface under the cumulative ranking (SUCRA) curve, together with a rankogram plot, to provide a hierarchy of treatments with consideration of both the location and the variance of all relative treatment effects. The efficacy of each intervention was expressed as a percentage, which was considered in relation to an imaginary intervention assumed to be the best. The higher the SUCRA value is, the higher the likelihood of effective treatment was. The ROB graphs were constructed by using Review Manager 5.4 software.

## 3. Results

### 3.1. Basic Characteristics of Included Studies

A total of 6131 articles were initially retrieved. After reviewing titles and abstracts, 236 studies progressed to full manuscript review. Of these, 213 full-texts were excluded and the remaining 23 individual studies involving 8877 participants were finally included [5,8,13,14,15,16,17,18,21,22,23,24,25,26,27,28,29,30,31,32,33,34,35] (Figure 1).

The characteristics of included studies were presented in Table 1 and Table 2. These trials were published between 2008 and 2021, and the sample size varied from 53 to 2153 participants. In general, recruited women were overweight (body mass index [BMI] ≥ 25 kg/m^2^) or obese (BMI ≥ 30 kg/m^2^), except for two Chinese studies enrolling participants with BMI ≥ 24 kg/m^2^ according to Chinese categories (overweight 24–27.9 kg/m^2^ and obese ≥ 28 kg/m^2^) [5]. Detailed inclusion and exclusion criteria in each study were summarized in Table 1. Considering that metformin was the sole medication used in included RCTs, the interventions are categorized into four groups: diet alone, PA alone, combined diet + PA intervention, and medication (i.e., metformin) (detailed information in Table 2).

Risk of bias summary and each risk of bias item presented as percentages across all included studies were displayed in Appendix A, respectively. All 23 trials had a low risk of selection bias by using random sequence generation, except for one trial which chose to use one opaque envelope. In 15 trials, the performance bias was at high risk. Nevertheless, these results were acceptable when considering that it is difficult to blind participants to group assignment in diet/PA intervention protocols. The detection bias was at high risk in five trials because of absent blinding of outcome assessors. High risk of attrition bias was observed in four trials because of their high risk of lost follow-up (>10%). The reporting bias was at high risk in two trials given that their primary outcomes were not completely reported. Other bias was at high risk in three trials due to imbalanced baseline characteristics between groups. In addition, by reason that relative descriptions were absent, the risks of selection bias, performance bias and detection bias were unclear in seven, four and five trials, respectively. Overall, the 23 articles were all judged to be of low ROB.

### 3.2. Network Meta-Analysis

Figure 2 illustrated the network maps of the studies examining the effectiveness of various interventions (i.e., diet, PA, diet + PA, and medication), and direct comparisons between these interventions. The size of the nodes reflected the sample size in the corresponding intervention type, and the edge thickness was proportional to the number of studies for that comparison. A contribution plot (Appendix A) was employed to evaluate the contribution of different direct comparisons to this NMA. The size of each square was proportional to the weight attached to each direct summary effect (horizontal axis) for the estimation of each network summary effects (vertical axis).

Primary Outcome. The network forest plot of Figure 3A showed the RRs (95% CIs) of all 22 individual direct pair comparisons grouped in seven regimen pairwise meta-analyses. Unexpectedly, although there was a trend that PA and diet + PA intervention were protective factors of GDM occurrence, all comparisons yielded insignificant results. Furthermore, the network forest plot of Figure 3B showed RRs (95% CIs and prediction intervals [PIs]) of all 10 direct and indirect comparisons in this NMA (seven direct and three indirect). Despite failure to reach significance, PA and die + PA intervention, especially PA intervention alone, seemed better than diet intervention alone (RR 0.75, 95%CI [0.50, 1.11]; RR 0.81, 95%CI [0.58, 1.13]). There was a tendency that the efficacy of medication (RR 1.02, 95%CI [0.73, 1.42]), less effective than PA (RR 1.24, 95%CI [0.78, 1.96]) and diet + PA intervention (RR 1.14, 95%CI [0.77, 1.67]), was similar to usual care, and diet intervention alone (RR 1.10, 95%CI [0.83, 1.46]) even played a negative effect in GDM prevention.

To determine the relatively optimal intervention for preventing GDM, a cumulative ranking probability plot (Figure 4A) and rankogram plot (Figure 4B) were employed for ranking these interventions. Among included trials, PA intervention had the highest probability of being the best strategy (SUCRA value of 82.8% compared with 70.7% for diet + PA intervention). The other two interventions (diet and medication) had lower SUCRA values. The SUCRA value of diet intervention alone (SUCRA, 19.7%) was even worse than usual care (SUCRA, 38.7%), thus becoming the least efficacious regimen. 

Secondary Outcome. Figure 5A showed the MD (95% CIs) of all 18 individual direct pair comparisons grouped in four regimen pairwise meta-analyses. Compared with usual care, women with PA, diet and combined diet + PA interventions all gained less weight. The difference of GWG between usual care and medication was insignificant. Figure 5B showed MD (95% CIs and PIs) of all 10 direct and indirect comparisons in this NMA (4 direct and 6 indirect). Diet (MD −1.95, 95%CI [−3.19, −0.71]), PA (MD −1.98, 95%CI [−3.50, −0.47]) and combined diet + PA (MD −1.21, 95%CI [−1.92, −0.50]) interventions were significant factors for GWG restriction when compared with usual care. The differences between other direct and indirect comparisons were insignificant.

Figure 6A,B were employed for ranking the effectiveness of these interventions on GWG. Among included trials, the SUCRA values of usual care, PA, medication, diet + PA and diet interventions were 97.2%, 62.3%, 50.3%, 20.5% and 19.7%, respectively, indicating that the probability of usual care to gain weight was the highest. Given that GWG was the outcome of this NMA, while the goal of our study was to identify ideal interventions to restrict GWG, we ranked these interventions backwards. Therefore, diet intervention alone was the best strategy to restrict GWG, closely followed by diet + PA intervention. Following medication, PA intervention became the least efficacious regimen.

### 3.3. Publication Bias and Data Consistency

The funnel plot (Appendix A) appeared symmetrical, meaning that there was no evidence of publication bias or small study effects. By observing the network connections regarding different parametric data, we noticed four closed loops on the incidence of GDM in this NMA, whilst no closed loop was observed on GWG. Therefore, only an inconsistency plot on the incidence of GDM was conducted to detect any significant data inconsistency in the main results. Appendix A demonstrated that there were no serious risks of inconsistency in the included studies. The point estimates of direct and indirect comparisons for GWG were evaluated. The evidence was downgraded by one level if the prediction intervals extended across the line of no effect.

## 4. Discussion

### 4.1. Main Findings

Our NMA demonstrated that neither diet, PA, combined diet + PA nor medication intervention could effectively protect overweight/obese women from developing GDM. Despite the discrepancy with other literature studying women without special BMI [11,37,38], this conclusion remains convincing due to consistency with studies targeting overweight/obese pregnant women [39,40]. There are some possible explanations for the insignificance. On the one hand, providing intervention only during gestation, which results in a short duration of these interventions, might not be sufficient to produce remarkable metabolic improvements and eventually prevent GDM. During healthy pregnancy, placental hormones and metabolic adaptations promote the state of insulin resistance (IR), which facilitates the adequate transfer of glucose to the fetus [39,41]. By reasons that the excess of adipose tissue affects the production of adipokines, chemokines and cytokines, IR is exaggerated in overweight/obese pregnant women [39], making them enter pregnancy with increased IR, which further deteriorates as pregnancy progresses [39]. When insulin secretion does not increase adequately to counterbalance the state of IR in the second half of pregnancy, maternal glucose intolerance appears and may contribute to the increased risk of GDM [42]. This emphasizes the significance of reducing BMI before conception. A large study including 226,958 women also indicated that a 10% reduction in pre-pregnant BMI reduced the risk of GDM by up to 25%, depending on the baseline BMI [43]. In this regard, intervention initiated at the onset of pregnancy or even the preconception period should be recommended among overweight/obese women to prevent GDM. On the other hand, the intensity of interventions, for instance, the dose of metformin administrated and the type of PA intervention (PA counselling vs. supervised exercise sessions, or diet counselling sessions vs. diet brochures), is heterogeneous in included trials. Given that overweight/obesity is an established risk factor for GDM [4], and is closely associated with metabolic disorders [44], participants with overweight/obesity may benefit less from modest interventions. Besides, overweight and obesity are intrinsically linked with long-term excessive calorie intake [41], which might be uneasy to change, resulting in a barrier to diet intervention. The fear of harming self and the baby, pregnancy-induced physical discomfort and discouragement by others [45] constitute barriers to PA intervention. Apprehension regarding the use of medications during pregnancy, for example, anxiety related to metformin crossing the placenta and reaching the fetus, also exists [46]. All of these might contribute to the poor compliance of overweight/obese women to the above interventions, which could be another reasonable factor for their failure to prevent GDM. Therefore, a dose-response phenomenon may be present, where more robust interventions, such as supervised exercise sessions, should be principally examined in the future.

More importantly, although we failed to reach significance on the efficacy of various interventions among overweight/obese women in this NMA, there was a tendency that PA and diet + PA interventions played protective roles in GDM development. Of the four included interventions, PA intervention ranked as the optimal strategy to prevent GDM, followed by diet + PA intervention and medication, which is partially supported by Davenport et al. [11]. PA has long been prescribed to patients with diabetes due to improvements in glycemia and insulin sensitivity. Studies have proposed that PA achieves these benefits by promoting the glucose uptake of skeletal muscle and increasing mitochondrial density as well as the expression of glucose transporter proteins [47]. Moreover, PA also improves pancreatic islet cell function, increases myonectin levels, and decreases adipokine levels and oxidative stress [48]. These results laid a solid foundation for the possible effectiveness of PA and PA-based lifestyle intervention on the prevention of GDM. Oral antidiabetics, especially metformin, are another empirically therapeutic strategy for women with polycystic ovary syndrome (PCOS) or obesity. Nevertheless, despite the fact that metformin has the ability to inhibit hepatic gluconeogenesis and glucose absorption, stimulate glucose uptake in peripheral tissues, reduce weight gain and improve insulin resistance [46,48], our NMA cannot provide evidence to support metformin as a feasible regime in overweight/obese women during pregnancy to prevent GDM, which is consistent with a previous Cochrane review [36]. Interestingly, we found that diet intervention alone was the least effective regimen, and this conclusion was supported by a meta-analysis pooling 20 RCTs, which also showed that nutritional manipulation in pregnancy cannot reduce the risk of GDM [49]. Overall, this finding suggested that diet intervention alone was unable to reverse the deteriorating state of IR and prevent GDM in overweight/obese women. However, despite the failure of all evaluated interventions to reach significance, given that the risk of developing GDM in overweight and obese pregnant women is twofold higher than in non-obese women [5], not providing any intervention for them is unreasonable. Consequently, this NMA indicates that although no specific behavioral or medication interventions were effective enough to prevent GDM in overweight/obese women, exercise or exercise combined with diet counseling could be recommended as relatively ideal strategies if necessary, while diet counseling alone should not be currently considered a viable treatment option for overweight/obese pregnant women to prevent GDM.

Excessive GWG is another significant risk factor for GDM [42]. It is worth noting that diet, PA and combined diet + PA interventions, instead of medication, were all protective factors for GWG restriction among overweight/obese pregnant women, and this conclusion is not completely consistent with a previous NMA [39] which suggested that both metformin and PA were effective. The possible reason is the different sample size included in the NMA. Fourteen studies with 2371 women were included in the above NMA when analyzing metformin and exercise, whilst 18 studies with 7018 women were included in our NMA, making our NMA more convincing. Because excessive GWG is tightly associated with increased risks of a variety of adverse maternal and offspring outcomes [7,8,9], the effectiveness ranking of interventions aimed at restricting GWG should be evaluated. In the current NMA, diet intervention was the best strategy to limit GWG, closely followed by diet + PA intervention, whereas PA intervention became the least efficacious regimen. This finding of our NMA could support the conclusion of a previous meta-analysis pooling 23 RCTs which indicated that a healthy eating strategy was likely to have a larger effect size than a combined healthy eating and PA strategy in restricting GWG [45]. Although not completely consistent with our NMA, this meta-analysis further emphasized the significance of diet and combined diet + PA interventions to restrict GWG. Moreover, the role of maternal behaviors, in particular reducing dietary fat and regular PA, are profoundly crucial in epigenetic programming of metabolic disease risk in offspring [40]. As a consequence, with a variety of benefits, diet intervention alone or combined diet + PA intervention can be considered as viable strategies for overweight or obese pregnant women to restrict GWG.

### 4.2. Strengths and Limitations

Our NMA is systematic and exhaustive, including 8877 participants and evaluating a wide range of interventions possibly applied during pregnancy. In addition, by adopting strict inclusion criteria (e.g., only including participants with overweight/obesity) to limit heterogeneity, the conclusions are precise and compelling. The lack of inconsistency further strengthens the results of this NMA.

Our study also has several limitations, however. In this NMA, participants were recruited at <20 weeks of gestation, resulting in a short intervention duration. However, in some of these studies, the interventions were initiated at early pregnancy, which may play a more significant role in the prevention of GDM, and we are not able to perform a subgroup analysis for the length of intervention. The races of participants and the diagnostic criteria of GDM are different among included studies, which are significant factors affecting the incidence of GDM [2], and might further slightly affect our conclusions. However, the characteristics of participants and the diagnostic criteria of GDM are homogeneous between the intervention group and the control group in each RCT, which mean that the impact of these differences on the effectiveness of interventions could be excluded. In addition, we only considered the incidence of GDM and GWG as outcomes of this NMA. In order to determine the effects of these interventions, possible adverse outcomes resulted from these interventions, such as intrauterine growth retardation and preterm delivery, which were not analyzed in this NMA. When evaluating GWG, the evidence was downgraded by one level, due to the lack of closed loops and that the PIs extended across the line of no effect. Furthermore, the benefit of any intervention needs to be balanced against the resources invested to achieve this consequence, and an economic evaluation should be performed to examine the cost-effectiveness of these interventions.

Moreover, an individual patient data network meta-analysis (IPD-NMA) is a useful evidence synthesis method to estimate the relative effectiveness of multiple competing interventions, which was not performed in our study. By pooling patient-level data across various studies, interactions between baseline individual characteristics and treatments could be examined with more power [50]. Furthermore, IPD-NMA enables researchers to control for more potential confounding factors [51], which would further improve the quality of an NMA. In addition, IPD-NMA could also explore outcomes in potentially important subgroups and identify the population that may benefit most from a specific intervention, which is severely limited in aggregated data meta-analyses [52]. Consequently, IPD-NMA should be encouraged in future studies to draw more precise conclusions. Meanwhile, compared with a frequentist framework, a Bayesian framework is sometimes more appropriate for smaller data sets, where there is a shift of emphasis from large-sample theory to specific probability distributions [53,54]. Nevertheless, the Bayesian methods were not utilized in our NMA and should be performed in the future, especially for further subgroup analysis, such as for gestational week at inclusion into studies and participants’ BMI.

## 5. Conclusions

Neither diet, PA, combined diet + PA intervention nor medication could offer significant benefit in preventing GDM in overweight/obese women, and renewed efforts are hence needed to seek new and effective interventions and to prevent obesity prior to conception. However, PA, diet and combined diet + PA interventions were all remarkably beneficial for GWG restriction. In aggregate, combined diet + PA intervention seemed the superior choice as a significant protective factor to restrict GWG, and it also had a tendency to prevent GDM occurrence.

## Figures and Tables

**Figure 1 nutrients-14-02383-f001:**
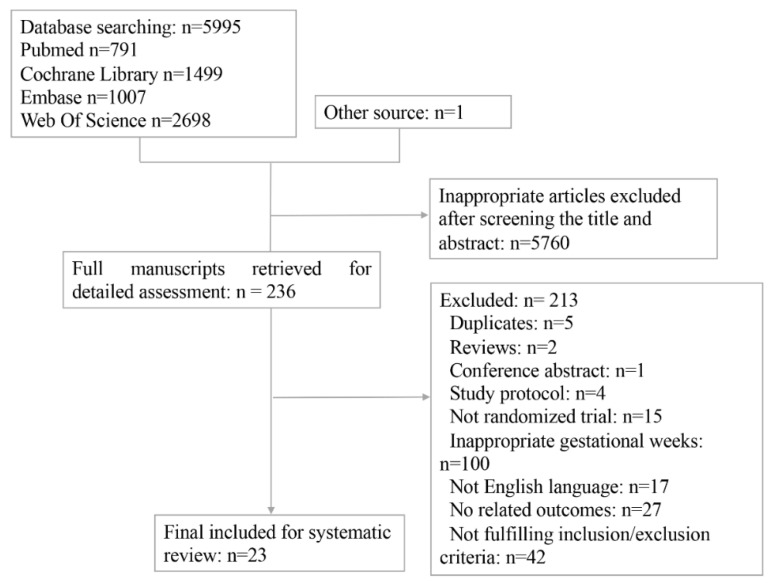
Flow chart of included studies in the network meta-analysis.

**Figure 2 nutrients-14-02383-f002:**
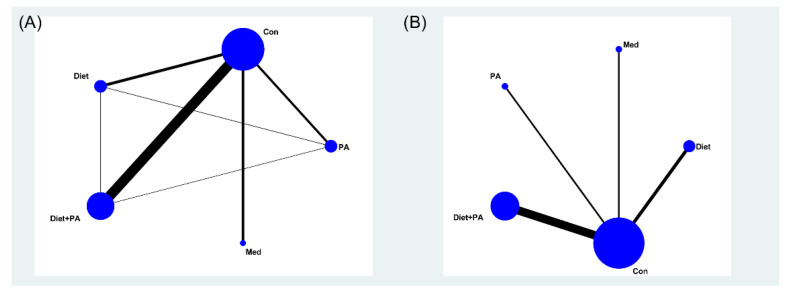
Network meta-analysis maps of the incidence of gestational diabetes mellitus (GDM) (**A**) and gestational weight gain (GWG) (**B**). Abbreviations: physical activity, PA; control, Con; medication, Med.

**Figure 3 nutrients-14-02383-f003:**
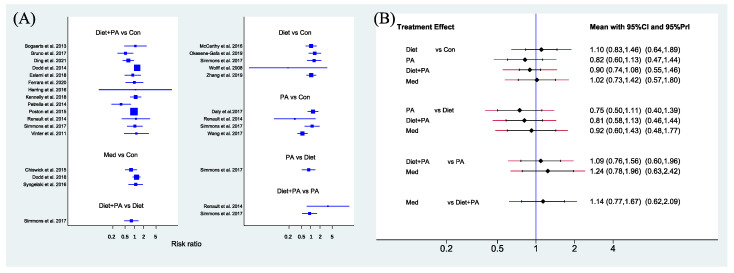
Network forest plot of direct pairwise comparisons of regimens (**A**), and of both direct and indirect comparisons of regimens (**B**) on the incidence of GDM. Abbreviations: physical activity, PA; control, Con; medication, Med [5,8,13,14,15,16,17,18,21,22,23,24,25,26,29,30,31,32,33,34,35,36].

**Figure 4 nutrients-14-02383-f004:**
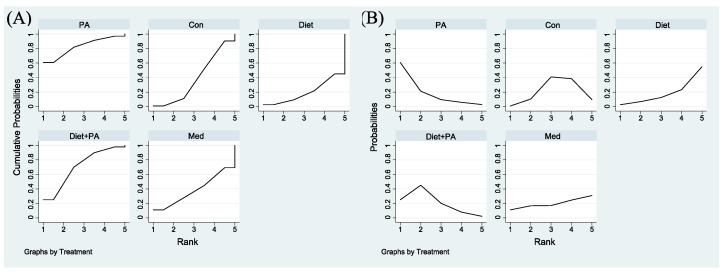
Surface Under the Cumulative Ranking for the incidence of GDM (**A**). Rankograms for the incidence of GDM (**B**) were derived from relevant surface under the cumulative ranking (SUCRA) values for various regimens. Abbreviations: physical activity, PA; control, Con; medication, Med.

**Figure 5 nutrients-14-02383-f005:**
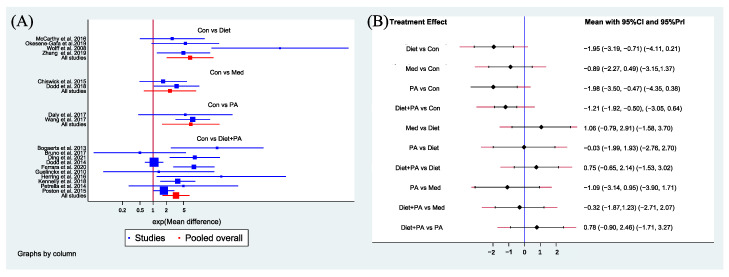
Network forest plot of direct pairwise comparisons of regimens (**A**), and of both direct and indirect comparisons of regimens (**B**) on GWG. Abbreviations: physical activity, PA; control, Con; medication, Med [5,8,13,14,15,16,17,21,23,24,25,27,29,31,32,33,34,36].

**Figure 6 nutrients-14-02383-f006:**
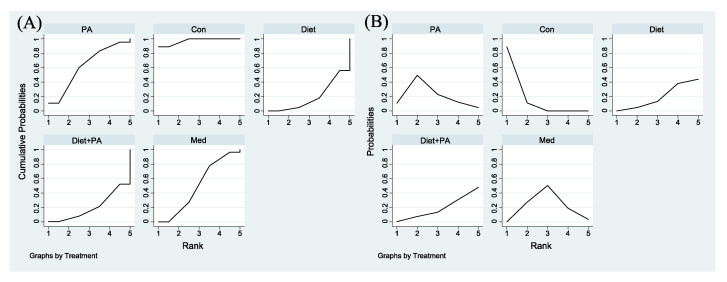
Surface Under the Cumulative Ranking for GWG (**A**). Rankograms for GWG (**B**) were derived from relevant SUCRA values for various regimens. Abbreviations: physical activity, PA; control, Con; medication, Med.

**Table 1 nutrients-14-02383-t001:** Characteristics of included studies.

Author (Year)	Conflict of Interest	Ethical Approval	Time Frame	Country	Inclusion Criteria	Exclusion Criteria	Sample Size
Bruno R. et al. (2017) [13]	None	Yes	2013–2014	Italy	Pre-pregnancy BMI ≥ 25 kg/m^2^; singleton pregnancy; 9–12 weeks of pregnancy	Chronic diseases, including diabetes mellitus, hypertension; medical conditions or dietary supplements that might affect body weight (i.e., thyroid diseases); previous bariatric surgery, contraindications to exercise; previous GDM and smoking habits	131
Poston L. et al. (2015) [31]	None	Yes	2009–2014	UK	Singleton pregnancy; BMI ≥ 30 kg/m^2^; 15^+0^–18^+6^ weeks’ gestation	Pre-pregnancy diagnosis of essential hypertension, diabetes, renal disease, SLE, antiphospholipid syndrome, sickle-cell disease, thalassaemia, coeliac disease, thyroid disease, current psychosis; currently prescribed metformin	1280
Okesene-Gafa K.A.M. et al. (2019) [29]	None	Yes	2015–2017	New Zealand	12^+0^–17^+6^ weeks of gestation; singleton pregnancy; BMI ≥ 30 kg/m^2^	Pre-existing diabetes or HbA1c ≥ 50 mmol/mol; taking capsules or supplements containing probiotics; previous bariatric surgery; severe hyperemesis; medications or medical conditions that alter glucose metabolism	196
Renault K.M et al. (2014) [30]	None	Yes	2009–2012	Denmark	BMI ≥ 30 kg/m^2^; singleton pregnancy; <16 weeks of gestation	Multiple pregnancy; pre-pregnant diabetes, serious diseases limiting PA; previous bariatric surgery; alcohol or drug abuse	362
Wang C. et al. (2017) [5]	None	Yes	2014–2016	China	Singleton pregnancy; pre-pregnancy BMI ≥ 24 kg/m^2^; <12^+6^ weeks’ gestation	Cervical insufficiency; pre-existing hypertension, diabetes, cardiac disease, renal disease, SLE, thyroid disease or psychosis; currently treated with metformin or corticosteroids	265
Ding B. et al. (2021) [17]	None	Yes	2015–2016	China	BMI ≥ 24 kg/m^2^; <12 weeks of gestation	Previous GDM and macrosomia; history of diabetes, PCOS, hyperthyroidism or hypothyroidism; threatened abortion	215
McCarthy E.A. et al. (2016) [25]	None	Yes	2011–2012	Australia	BMI ≥ 25 kg/m^2^; <20 weeks of gestation; singleton pregnancy	Pre-existing diabetes	366
Simmons D. et al. (2017) [35]	NR	Yes	2012–2015	the UK, Ireland, Netherlands, Austria, Poland, Italy, Spain, Denmark, Belgium	Pre-pregnancy BMI ≥ 29 kg/m^2^; ≤19^+6^ weeks of gestation; singleton pregnancy	Diagnosis of GDM; preexisting diabetes; chronic medical conditions (e.g., valvular heart disease) or a psychiatric disorder; inability to walk ≥ 100 m safely; requirement for a complex diet	397
Vinter C. A. et al. (2011) [22]	None	Yes	2007–2010	Denmark	10–14 weeks of gestation; pre-pregnancy BMI of 30–45 kg/m^2^	Prior serious obstetric complications; major medical disorders; positive OGTT in early pregnancy; alcohol or drug abuse; multiple pregnancy.	304
Wolff S. et al. (2008) [16]	NR	Yes	NR	Denmark	Nondiabetic; BMI ≥ 30 kg/m^2^; in their early pregnancy (15 ± 3 weeks of gestation)	Smoking; multiple pregnancy; medical complications known to adversely affect fetal growth or to contraindicate limitation of weight gain	53
Bogaerts A.F. et al. (2013) [21]	None	Yes	2008–2011	Belgium	Pre-pregnancy BMI ≥ 29 kg/m^2^; <15 weeks of gestation	>15 gestational week; pre-existing T1DM; multiple pregnancy; primary need for nutritional advice	197
Zhang Y. et al. (2019) [34]	None	Yes	2012–2015	China	≤16 weeks of gestation; BMI ≥ 24 kg/m^2^	Artificial impregnation; history of hypertension, diabetes, or coronary heart disease; mental disorder; special dietary needs	400
Petrella E. et al. (2014) [24]	None	Yes	2011	Italy	Pre-pregnancy BMI ≥ 25 kg/m^2^; singleton pregnancy; during their 12th week of gestation	Twin pregnancy; chronic diseases; previous history of GDM; smoking during pregnancy; previous bariatric surgery; engaging in regular PA; dietary supplements or herbal products known to affect body weight	61
Chiswick C. et al. (2015) [23]	None	Yes	2011–2014	The UK	BMI ≥ 30 kg/m^2^; 12–16 weeks’ gestation	Pre-existing diabetes; previous or current diagnosis of GDM; systemic disease; previous delivery of a baby < the 3rd percentile for weight; previous pregnancy with PE prompting delivery before 32 weeks’ gestation; known hypersensitivity to metformin; known liver/renal failure; acute disorders with the potential to change renal function; lactation; multiple pregnancy	295
Dodd J.M. et al. (2018) [36]	None	Yes	2013–2016	Australia	Singleton pregnancy; 10–20 weeks’ gestation; BMI ≥ 25 kg/m^2^	Multiple pregnancy; pre-pregnant T1DM or T2DM; significant renal or hepatic impairment	514
Dodd J.M. et al. (2014) [14]	None	Yes	2008–2011	Australia	BMI ≥ 25 kg/m^2^; singleton pregnancy; 10^+0^–20^+0^ weeks’ gestation	Pre-pregnant T1DM or T2DM; multiple pregnancy	2153
Ferrara A. et al. (2020) [8]	None	Yes	2014–2017	The U.S	Pre-pregnancy BMI of 25.0–40.0 kg/m^2^; singleton pregnancy	Fertility-assisted pregnancy; bed rest; diagnosis of (gestational) diabetes; current uncontrolled hypertension; thyroid disease diagnosed in last 30 days; history of cardiovascular, cancer, lung or serious gastrointestinal disease; history of eating disorder of bariatric surgery; serious mental illness; recent history of mood or anxiety disorder; drug or alcohol use disorder; >13 weeks’ gestation	389
Syngelaki A. et al. (2016) [18]	None	Yes	2010–2015	The UK	Without diabetes; BMI > 35 kg/m^2^; 12–18 weeks of gestation; singleton pregnancy	Previous history of GDM; kidney, liver or heart failure; serious medical condition; hyperemesis gravidarum; treatment with metformin at the time of screening; known sensitivity to metformin	397
Daly N. et al. (2017) [33]	None	Yes	2013–2016	Ireland	BMI ≥ 30 kg/m^2^; <17 weeks of gestation	Multiple pregnancy; pre-existing diabetes; hypertension; alcohol or drug abuse; medication affecting insulin secretion or sensitivity; serious cardiorespiratory disorders; hepatic or renal impairment; SLE; hematologic disorders; celiac disease; thyroid disorders; current psychosis; malignant disease	86
Kennelly M.A. et al. (2018) [32]	None	Yes	2013–2016	Ireland	Singleton pregnancy; 10–15 weeks of gestation; BMI of 25.0–39.9 kg/m^2^	multiple pregnancy; medical disorder requiring treatment; previous history of GDM or previous poor obstetric outcome	498
Eslami E. et al. (2018) [26]	None	Yes	2016–2017	Iran	16–20 weeks of gestation; singleton pregnancy without complications; BMI > 25 kg/m^2^ in the first trimester; the pregnancy being the female’s first, second or third	Physical or mental illness; maternal diabetes; history of hospitalization in the current pregnancy; at risk of preterm delivery; addiction or habitual use of drugs and alcohol; history of infertility and the use of ART	140
Herring S.J. et al. (2016) [15]	None	Yes	2013–2014	The U.S	<20 gestational week; BMI 25–45 kg/m^2^ at first trimester; medicaid recipient	Conditions requiring specialized nutritional care; endorsed current tobacco use; multiple pregnancy	56
Guelinckx I. et al. (2010) [27]	None	Yes	2006–2008	Belgium	BMI > 29 kg/m^2^; <15 weeks of gestation	Pre-existing diabetes or developing GDM; multiple pregnancy; >15 weeks of gestation; premature labor; primary need for nutritional advice in case of a metabolic disorder; kidney problems; Crohn’s disease; allergic conditions	122

Abbreviations: not reported, NR; body mass index, BMI; gestational diabetes mellitus, GDM; systemic lupus erythematosus, SLE; polycystic ovarian syndrome, PCOS; oral glucose tolerance test, OGTT; fasting blood glucose, FBG; glycosylated hemoglobin, HbA1c; type 1 diabetes mellitus, T1DM; type 2 diabetes mellitus, T2DM; pre-eclampsia, PE; physical activity, PA; in-vitro fertilization, IVF; assisted reproductive technology, ART; small for gestational age, SGA; β-human chorionic gonadotropin, β-hCG.

**Table 2 nutrients-14-02383-t002:** Protocol and outcomes definition in the included studies.

Author (Year)	Intervention	Intervention Time	Comparison	Outcomes Reported
Bruno R. et al. (2017) [13]	Lifestyle intervention (consisting of individualized counselling with the prescription of a hypocaloric, low-glycaemic and low-fat diet associated with PA recommendations and a close follow-up)	NR	Standard dietary recommendations	GDM, GWG
Poston L. et al. (2015) [31]	Health trainer-led sessions related to behavioural changes; dietary and PA advices	Initiated within one week of randomization and lasted for eight weeks	Routine antenatal care	GDM, GWG
Okesene-Gafa K.A.M. et al. (2019) [29]	A handbook about healthy eating; home-based education sessions about diet, weight gain and behavioural changes; dietary intervention visits; motivational text messages	Four dietary sessions aimed to be completed before 26–28 weeks’ gestation; text messages from randomization until birth	Routine dietary advice	GDM, GWG
Renault K.M. et al. (2014) [30]	PA group: individually advised and encouraged to increase PA PA + D group: individually advised and encouraged to increase PA and dietary contacts with the dietitian	Immediately after randomization	Usual standard regimen for obese pregnant women	GDM
Wang C. et al. (2017) [5]	Supervised cycling exercises	Initiated within three days of randomization until weeks 36–37	Usual daily activities	GDM, GWG
Ding B. et al. (2021) [17]	Personalized dietary and exercise sessions, and online monitoring to promote adherence	NR	A general session about nutrition and weight management	GDM, GWG
McCarthy E.A. et al. (2016) [25]	Targeted, serial self-weighing and simple dietary advice	NR	Standard care	GDM, GWG
Simmons D. et al. (2017) [35]	Individual lifestyle coach to promote a lower carbohydrate, lower fat, higher fiber and higher protein diet, and/or both aerobic and resistance physical activity, using face-to-face sessions and telephone calls or E-mails; recommendation for a limitation in GWG to 5 kg	At least four face-to-face coaching sessions before 24–28 weeks, and all completed by 35 weeks of gestation.	Usual care	GDM
Vinter C. A. et al. (2011) [22]	Lifestyle intervention (including diet counselling sessions to limit GWG, encouragement for PA and access to supervised training classes)	Dietary counselling at 15, 20, 28 and 35 weeks’ gestation	Access to a website with advice about dietary habits and PA	GDM
Wolff S. et al. (2008) [16]	Dietary consultations for healthy eating and weight gain management	NR	No dietary consultations	GDM, GWG
Bogaerts A.F. et al. (2013) [21]	Brochure group: brochure about nutritional advice and PA to limit GWG Prenatal session group: the brochure and prenatal lifestyle intervention sessions mainly focusing on healthy energy intake, exercise and motivation	Sessions before 15 weeks of gestation and between 18–22, 24–28 and 30–34 weeks of gestation	Routine antenatal care	GDM, GWG
Zhang Y. et al. (2019) [34]	Individualized dietary GI and GL assessment using an app, and instructions to achieve low GI diet	NR	Standard nutrition consultation	GDM, GWG
Petrella E. et al. (2014) [24]	Lifestyle intervention (including dietary counseling sessions and advices for moderate PA)	NR	A simple nutritional booklet about lifestyle	GDM, GWG
Chiswick C. et al. (2015) [23]	Metformin 2500 mg daily	From 12–16 weeks’ gestation until delivery	Placebo	GDM, GWG
Dodd J.M. et al. (2018) [36]	Metformin 2000 mg daily	From randomization until delivery	Placebo	GDM, GWG
Dodd J.M. et al. (2014) [14]	A comprehensive dietary and lifestyle intervention which included dietary, exercise, and behavioral advices	From within two weeks of randomization until 36 weeks’ gestation	Standard care	GDM, GWG
Ferrara A. et al. (2020) [8]	A lifestyle intervention sessions behavior strategies to improve weight, diet, PA and stress management	Until 38 weeks’ gestation	Usual care	GDM, GWG
Syngelaki A. et al. (2016) [18]	Metformin 3.0 g daily	From 12–18 weeks’ gestation until delivery	Placebo	GDM
Daly N. et al. (2017) [33]	Supervised exercise classes	For the duration of their pregnancy and for up to six weeks postpartum	Routine prenatal care	GDM, GWG
Kennelly M.A. et al. (2018) [32]	Lifestyle intervention (including specific dietary and exercise advice) mainly supported by a smartphone application	From randomization until delivery	Standard antenatal care	GDM, GWG
Eslami E. et al. (2018) [26]	Lifestyle intervention lectures, a booklet and educational text messages on nutrition and PA advices	From week 16–20 to week 24–28	Routine pregnancy care	GDM
Herring S.J. et al. (2016) [15]	Technology-based behavioral intervention (including empirically-supported behavior change goals, interactive self-monitoring text messages, health coach calls and online skills training and support)	From baseline until delivery	Standard obstetrical care	GDM, GWG
Guelinckx I. et al. (2010) [27]	Passive group: a brochure about nutrition, PA and weight gain management Active group: the brochure and lifestyle education by a nutritionist	NR	Routine prenatal care	GWG

Abbreviations: not reported, NR; body mass index, BMI; gestational diabetes mellitus, GDM; gestational weight gain, GWG; physical activity, PA; glycemic index, GI; glycemic load, GL.

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
