# Peer review of "Prevention of Gestational Diabetes Mellitus and Gestational Weight Gain Restriction in Overweight/Obese Pregnant Women: A Systematic Review and Network Meta-Analysis"

_nutrients, 2022, doi:10.3390/nu14122383_

Round 1

Reviewer 1 Report

dear colleagues,

thank you for the interesting review.

I suggest that you add one additional paragraph in the discussion about the importance of doing a ipd data meta analysis as this may reveal other information + the importance of doing bayesian meta analyses after this nma because subgroups need to get further attention.

Author Response

Point 1: I suggest that you add one additional paragraph in the discussion about the importance of doing a ipd data meta analysis as this may reveal other information + the importance of doing bayesian meta analyses after this nma because subgroups need to get further attention.

Response 1: 

Thank you for your kind suggestion, which significantly improved the quality of our study. We have added one additional paragraph in the ‘strengths and limitations’ section about the importance of doing both an IPD meta-analysis and Bayesian meta-analyses.

We now have stated as follows (please see line 393-409 on page 15): ‘Moreover, individual patient data network meta-analysis (IPD-NMA) is a useful evidence synthesis method to estimate the relative effectiveness of multiple competing interventions, which was not performed in our study. By pooling patient-level data across various studies, interactions between baseline individual characteristics and treatments could be examined with more power [50]. Besides, IPD-NMA enables us to control for more potential confounding factors [51], which would further improve the quality of the studies. In addition, IPD-NMA could also explore outcomes in potentially important subgroups and identify the population who may benefit most from a specific intervention, which is severely limited in aggregated data meta-analyses [52]. Consequently, IPD-NMA should be encouraged in future studies to draw more precise conclusions. Meanwhile, compared with a frequentist framework, a Bayesian framework is sometimes more appropriate for smaller data sets, where there is a shift of emphasis from large-sample theory to specific probability distributions [53, 54]. Nevertheless, the Bayesian methods were not utilized in our NMA, and should be performed in the future, especially for further subgroup analysis which needs to get more attention. Subgroup analyses focusing on gestational week at the beginning of intervention and participants’ BMI could be more promising’.

Reviewer 2 Report

In this network meta-analysis, the authors aimed to assess the comparative efficacy of different interventions during pregnancy on preventing GDM and restricting GWG among overweight/obese women. The paper is interesting and the conclusions are supported by results,  I only have minor concerns:

1)    The definition of GDM must be updated (see ADA 2022)

2)    It must be precised that prevalence of GDM is very different among countries. The difference origins from several factors, including different screening criteria.

3)    In Materials and Methods section, please confirm that the electronic searching was also supplemented by retrieving additional studies from its references and citations and from the PubMed option ‘Related Articles’.

4)    About point 2), can the authors exclude that these differences affected their analyses?

Author Response

Point 1: The definition of GDM must be updated (see ADA 2022).

Response 1:Thank you for your important suggestion. We have updated the definition of GDM according to the latest ADA guidelines in 2022. We now have stated as follows (please see line 33-37 on page 1): ‘Gestational diabetes mellitus (GDM) has been defined as glucose intolerance with onset or first recognition during pregnancy for many years, and now has been found with serious limitations due to many cases of GDM representing preexisting hyperglycemia [1]. Nowadays, the latest definition of GDM should exclude women found to have diabetes by diagnostic criteria applied outside of pregnancy [1]’.

Point 2: It must be precised that prevalence of GDM is very different among countries. The difference origins from several factors, including different screening criteria.

Response 2: Thank you for your kind suggestion. We have revised the description of GDM prevalence as follows (please see line 37-39 on page 1): ‘GDM is one of the most common obstetric complications, with the prevalence varying from 7.5% to 27.0% among different areas, principally depending on different races and diagnostic criteria [2]’.

Point 3: In Materials and Methods section, please confirm that the electronic searching was also supplemented by retrieving additional studies from its references and citations and from the PubMed option ‘Related Articles’.

Response 3: Thank you for your kind advice. We added relative statements about the detailed search strategies in ‘materials and methods’ section, which was applied in our network meta-analysis but was ignored when writing. We now have added as follows (please see line 91-93 on page 2): ‘Detailed search strategies were listed in Appendix A, and supplemented with manual searches of the reference lists of included studies and relevant reviews as well as the PubMed option ‘Related Articles’’.

Point 4: About point 2), can the authors exclude that these differences affected their analyses?

Response 4: Thank you for your precise suggestion, which significantly improved the quality of our study. Our method to exclude these differences was to add additional discussion in the ‘strengths and limitations’ section as follows (please see line 379-384 on page 14): ‘The races of participants and the diagnostic criteria of GDM are different among included studies, which are significant factors affecting the incidence of GDM [2] and might result in our conclusion less powerful. However, the characteristics of participants and the diagnostic criteria of GDM are homogeneous between intervention group and control group in each RCT, which meant the impacts of these differences on the effectiveness of interventions could be excluded’.